# The *Aspergillus flavus hacA* Gene in the Unfolded Protein Response Pathway Is a Candidate Target for Host-Induced Gene Silencing

**DOI:** 10.3390/jof10100719

**Published:** 2024-10-16

**Authors:** Perng-Kuang Chang

**Affiliations:** Southern Regional Research Center, Agricultural Research Service, U.S. Department of Agriculture, 1100 Allen Toussaint Boulevard, New Orleans, LA 70124, USA; perngkuang.chang@usda.gov

**Keywords:** *Aspergillus flavus*, aflatoxin, unfolded protein response, bZip, CRISPR, host-induced gene silencing, RNAi, sclerotia, secondary metabolism

## Abstract

Fungal HacA/Hac1 transcription factors play a crucial role in regulating the unfolded protein response (UPR). The UPR helps cells to maintain endoplasmic reticulum (ER) protein homeostasis, which is critical for growth, development, and virulence. The *Aspergillus flavus hacA* gene encodes a domain rich in basic and acidic amino acids (Bsc) and a basic leucine zipper (bZip) domain, and features a non-conventional intron (Nt20). In this study, CRISPR/Cas9 was utilized to dissect the Bsc-coding, bZip-coding, and Nt20 sequences to elucidate the relationship between genotype and phenotype. In the Bsc and bZip experimental sets, all observed mutations in both coding sequences were in frame, suggesting that out-of-frame mutations are lethal. The survival rate of transformants in the Nt20 experiment set was low, at approximately 7%. Mutations in the intron primarily consisted of out-of-frame insertions and deletions. In addition to the wild-type-like conidial morphology, the mutants exhibited varied colony morphologies, including sclerotial, mixed (conidial and sclerotial), and mycelial morphologies. An ER stress test using dithiothreitol revealed that the sclerotial and mycelial mutants were much more sensitive than the conidial mutants. Additionally, the mycelial mutants were unable to produce aflatoxin but still produced aspergillic acid and kojic acid. RNAi experiments targeting the region encompassing Bsc and bZip indicated that transformant survival rates generally decreased, with a small number of transformants displaying phenotypic changes. Defects in the *hacA* gene at the DNA and transcript levels affected the survival, growth, and development of *A. flavus*. Thus, this gene may serve as a promising target for future host-induced gene-silencing strategies aimed at controlling infection and reducing aflatoxin contamination in crops.

## 1. Introduction

The endoplasmic reticulum (ER) is the organelle primarily responsible for protein folding and maturation. The unfolded protein response (UPR) is a vital mechanism that regulates cellular protein homeostasis, which is critical for cellular development and environmental adaptation [1]. The UPR has been implicated in human diseases such as cancer, diabetes, obesity, and neurodegeneration [2]. This mechanism is not activated under normal conditions; it is triggered by the abnormal accumulation of unfolded or misfolded proteins in the ER, which causes cellular stress. The UPR functions to restore protein folding balance to normal levels through multiple signaling pathways. Among these, the IRE1 pathway has been the best studied. Central to this pathway is a basic leucine zipper (bZip) transcription factor, XBP-1 (X-box-binding protein) in mammals and Hac1 in yeast, both of which have been extensively studied [3,4]. These transcription factors bind to a consensus UPR element in the promoters of ER-stress responsive genes to activate their expression [5,6].

The UPR is crucial for the growth, virulence, and antifungal susceptibility of *Aspergillus fumigatus* [7]. It has a major role in the pathogenicity of the necrotrophic fungus *Alternaria brassicicola* by altering its protection against host defense metabolites and reducing nutrient assimilation in the host environment [8]. In *Ustilago maydis*, the UPR regulates pathogenic development [9,10]. Fungal *hacA/hac1* genes, like those of yeast and mammalian cells, are regulated at the level of mRNA splicing [11]. The *hacA/hac1* mRNAs are spliced by IRE1, a kinase-nuclease protein, through a non-spliceosomal mechanism that removes a small non-conventional intron lacking the consensus splicing sequence “GU” at the 5′ splice site and “AG” at the 3′ splice site. In *Saccharomyces cerevisiae*, the fine tuning of the UPR is modulated by the chromatin remodeler Isw1, which binds to the ER stress-induced *hac1* mRNA, limiting the export of the resulting ribonucleoprotein from the nucleus [12]. Given its essential role in modulating the *hacA/hac1* mRNA levels, natural products and therapeutic inhibitors targeting the effector protein IRE1 in the UPR pathway have been proposed as alternative strategies for crop protection and the treatment of human blood cancer [13,14].

Disruption of the yeast *hac1* gene is not lethal but results in various effects on stress tolerance, mycelial growth, and cell wall integrity [15,16,17,18]. In fungi, the outcomes of *hac1*/*hacA* disruption vary by species. In *Neurospora crassa* and *Trichophyton rubrum*, the growth of the ∆*hac1*/∆*hacA* mutants is normal under non-stressed conditions, showing only slight sensitivity to cell wall perturbing agents such as Congo red and calcofluor white [19,20]. In *A. fumigatus*, deletion of *hacA* does not affect vegetative growth (Richi, 2009). In contrast, ∆*hacA* mutants of *Aspergillus niger* [21] and *Aspergillus oryzae* [22] exhibit severely retarded growth. An earlier study on *Aspergillus flavus* indicated that only mutants containing heterogeneous (wild-type and ∆*hacA*) nuclei were viable, although a more recent study argued otherwise [23,24]. Notably, a subsequent study on *N. crassa hac1* reported that the growth of the ∆*hac1* mutant was severely impaired [25].

*Aspergillus flavus* is a plant pathogen that infects corn, cottonseed, peanut, and tree nuts. It produces highly toxic and carcinogenic aflatoxins, particularly aflatoxin B_1_, which is classified as a Group 1 carcinogen by the International Agency for Research on Cancer [26]. To reduce pre-harvest colonization by toxigenic *A. flavus* and subsequent aflatoxin contamination, three major intervention technologies are currently employed: good agricultural practices, crop resistance breeding, and biological control [27]. The utilization of RNA interference (RNAi) mechanism involving microRNA (miRNA) has revolutionized plant research, especially in crop resistance to fungal pathogens [28]. Both small interfering RNA (siRNA) and miRNA regulate gene expression. The primary difference between them is that siRNA inhibits translation of a specific mRNA, while miRNA inhibits translation and induces cleavage of multiple mRNAs through partially complementary pairing to targeted mRNAs. siRNA originates from exogenous double-stranded RNA, such as that from an invading virus, whereas miRNA is single-stranded RNA derived from endogenous non-coding RNA that possesses a hairpin structure [29,30]. This host-induced gene silencing (HIGS) approach has been applied to *A. flavus* to reduce preharvest aflatoxin contamination. Single and multiplex HIGS strategies targeting aflatoxin biosynthesis pathway genes and, in some cases, additional developmental genes, have been reported to reduce aflatoxin accumulation in maize and peanuts under greenhouse or field conditions [31,32,33,34,35].

CRISPR/Cas9 (clustered regularly interspaced short palindromic repeats/CRISPR-associated nuclease) is a powerful technique for editing genome sequences [36]. It has been extensively used to study and improve gene functions across various biological systems. CRISPR/Cas9-derived mutants not involving a targeted repair sequence commonly exhibit random sequence changes, including deletions or insertions ranging from a few up to several thousand base pairs. These outcomes primarily arise because the excised regions are repaired by the non-homologous end-joining mechanism [37]. Many modified versions of CRISPR/Cas9 have been developed for use in *Aspergillus* species [38]. In specific applications, donor DNA can be included to create predefined sequence alterations, and deletion of large chromosomal segments, up to 300 kilobase pairs, has also been achieved [39,40].

In the present study, CRISPR/Cas9 was employed to investigate how sequence defects in the *A. flavus hacA* gene affected growth, development, and production of secondary metabolites. Three major phenotypes, conidial, sclerotial, and mycelial mutants, were obtained. These mutants exhibited different levels of sensitivity to dithiothreitol, an ER stress inducer. The mycelial mutants were unable to produce aflatoxin but still produced aspergillic acid and kojic acid, similar to the other mutant types. RNAi experiments targeting the *hacA* gene in *A. flavus* showed a general trend of decreased numbers of transformants regenerated from protoplasts.

## 2. Materials and Methods

### 2.1. Fungal Strains, Media, and Culturing Conditions

Wild-type *A. flavus* CA14 (SRRC 1708), an L-morphotype isolate known for producing abundant spores, along with derived *hacA* mutant strains were used in this study. For spore production, V8 juice agar medium was prepared using 50 mL V8 juice (Campbell Soup Company, Charlotte, NC, USA) and 20 g agar per liter, adjusted to pH 5.2 prior to autoclaving. Potato dextrose agar (PDA; EMD, Darmstadt, Germany) was used to characterize culture morphologies, specifically conidiation and sclerotial production. Both PDA and Czapek Solution Agar (CZ; Becton Dickinson, Sparks, MD, USA) without and with 10 mM dithiothreitol (DTT) were employed to investigate ER stress on growth and development. *Aspergillus* differential medium (ADM) [41] was used to examine production of aspergillic acid; it consisted of 1.5% tryptone, 1.0% yeast extract, 0.05% ferric citrate, and 1.5% agar. The occurrence of orange-colored ferriaspergillin, a complex of Fe^3+^ and three aspergillic acid molecules, indicated aspergillic acid production [42]. Coconut agar medium (CAM) [43] was used to assess production of anthraquinones, which are believed to be intermediates in the aflatoxin pathway. Commercial coconut milk (Goya Foods Inc., Secaucus, NJ, USA) purchased from a local Walmart supermarket was diluted with water (1:4), and 1.5% agar was added. The medium without adjustment of pH was autoclaved at 121 °C for 15 min. Kojic acid medium (KAM) used for kojic acid production [44] was slightly modified and contained 0.25% yeast extract, 0.1% K_2_HPO_4_·3H_2_O, 0.05% MgSO_4_·7H_2_O, 2% glucose, and 1.5% agar (pH 6.0). The production of kojic acid on KAM supplemented with 1 mM ferric chloride was indicated by the formation of diffusive, bright orange-red pigment, a chelate of kojic acid and Fe^+3^.

### 2.2. Preparation of CRISP/Cas9 Constructs

The protocol for constructing *A. flavus* CRISPR/Cas9 vectors has been described in detail [45]. Briefly, Addgene plasmid 191015 served as the template for PCR amplification of DNA fragments expressing gene-specific sgRNA cassettes. Each expression cassette contained the *A. flavus* U6 promoter, a short crRNA sequence, i.e., the protospacer sequence, fused to the scaffold tracrRNA sequence, and the *A. flavus* U6 terminator. Addgene plasmid 191016 served as the cloning vector, containing a codon-optimized *cas9* gene and half of the *AMA1* autonomously replicating sequence, with two unique restriction sites, *Pst*I and *Kpn*I, for cloning. To construct a CRISPR vector, two protospacer-specific PCR fragments were generated using primer sets of U6-F-P/gdRC_gene and gdFd_gene/U6-R-K, respectively (Table 1). The two PCR fragments without purification were directly fused in another round of PCR using primers U6-F-P and U6-R-K. The resulting fragment was cloned into the *Pst*I and *Kpn*I sites of plasmid 191016 using standard recombinant DNA methods.

### 2.3. Prediction of 3-Dimensional Structures

The *A. flavus* NRRL3357 *hacA* gene (AFLA_009139 = delisted AFLA_089270) is identical to that in other strains, including CA14 used in this study (data not shown). Therefore, the *hacA* gene annotation from the FungiDB database (version 3; https://fungidb.org/fungidb/app/; accessed on 28 February 2023) was utilized. The manually generated *A. flavus* HacA^∆^ amino acid sequence was used in a BlastP search of the NCBI database to identify a potential proxy protein annotated for *A. oryzae* genes. The result indicated that *A. flavus* HacA^∆^ shared 100% amino acid sequence identity with the annotated *A. oryzae* HacA. Consequently, *A. oryzae* HacA was used as the proxy for *A. flavus* HacA^∆^. The 3-dimensional models of HacA and HacA^∆^ proteins were retrieved from UniProKB (https://www.uniprot.org/uniprotkb/; accessed on 28 February 2023) for visual comparison.

### 2.4. Construction of hacA RNAi Vectors

The initial cloning vector was a shortened pPTRII from which a 3.0-kb *Pst*I fragment was removed and then blunt ended at 72 °C for 10 min using AccuPrime™ *Pfx* DNA polymerase (Invitrogen, Carlsbad, CA, USA). A *Hind*III-tagged DNA fragment containing the *A. flavus* U6 promoter and terminator sequences [45], with three restriction sites (*Pst*I, *Sma*I, and *Kpn*I) in between, was synthesized by Integrated DNA Technologies (Coralville, IA, USA) (Appendix A). The fragment was digested with *Hind*III and cloned into the linearized vector, which had also been digested with *Hind*III. The resulting vector was named V70-U6. For the hairpin RNAi construct, the loop portion was derived from a 52-bp intron of *A. flavus* AFLA_007258 (Appendix A), with minor modification. This gene is identical to the delisted AFLA_113120, which is highly expressed and encodes a glycosylphosphatidylinositol-anchored protein (GPI-AP) [46]. The stem portion consisted of 250 bps of the *A. flavus hacA* sequence from nucleotides 141 to 390 (Appendix A). Two *hacA* RNAi vectors were constructed. For the first vector, two PCR fragments amplified using primer pairs, StemR_K/LoopF and StemF_P/LoopR (Appendix A), were sequentially cloned into the *Sma*I/*Kpn*I and *Pst*1/*Sma*I sites of the V70-U6 vector. This *hacA* RNAi construct contained a stem of 162 bps and a loop of 103 bps, due to the presence of a *Sma*I site in the originally intended stem of 250 bps. The vector containing the right-side partial loop and the reverse complimentary sequence of the 250 bp portion was named R_bZip and served as a control. Similarly, for the second vector, two PCR fragments amplified using primer pairs, StemF_P/StemLp_BgII and StemR_K/LpStem_BgII, were sequentially cloned. The vector containing the left-side partial loop and the sequence of the 250 bp portion was named L_bZip and served as another control. The sequences of both vectors were confirmed through commercial whole plasmid sequencing at Plasmidsaurus (Louisville, KY, USA).

### 2.5. Generation of Protoplasts and Fungal Transformation

Approximately 3 × 10^7^ conidia were inoculated into 100 mL Czapek–Dox broth (Becton Dickinson, Sparks, MD, USA) supplemented with 0.5% Casamino Acids. The culture was shaken at 150 rpm for 11 to 12 h at 30 °C. The resulting mycelia were harvested using a 100-μm nylon cell strainer and resuspended in 20 mL of a filter-sterilized enzyme solution containing 1.0 g VinoTaste Pro (Novozymes, Bagsvaerd, Denmark) in a protoplasting solution of 0.55 M KCl, 0.05 M citric acid, pH 5.8. After approximately 3 h at 30 °C with shaking (60 rpm), protoplasts were collected by filtering through a 40 μm nylon cell strainer and pelleted using an IEC HN-SII table centrifuge at full speed for 10 min. The protoplasts were washed twice with a solution of 0.6 M KCl, 50 mM CaC1_2_, and 10 mM Tris-HCl, pH 7.5. Polyethylene glycol (PEG)–CaCl_2_-mediated transformation was performed as previously described [47]. The PEG solution consisted of 30% (*wt*/*vol*) PEG3350 (Rigaku, Japan), 0.6 M KCl, 50 mM CaCl_2_, and 10 mM Tris-HCl, pH 7.5. Typically, 2 × 10^6^ protoplasts and 0.25 μg of CRISPR vector DNA were used for each transformation for the study of relationships between genotype and phenotype. Transformants were selected on CZ regeneration plates supplemented with 0.1 µg/mL pyrithiamine. For RNAi experiments, the amounts of protoplasts and *Hind*III linearized vectors used were as specified in the text and related figure legends or table footnotes. A RNAi vector prepared from two independent *E. coli* clones was used for duplicate transformation. Small pyrithiamine-resistant colonies began forming within 2 to 3 days at 30 °C. Therefore, the total number of transformants was visually estimated on day 4, before growing colonies significantly overlapped or merged. Accurate determination of the total number of transformants was carried out under an Olympus SZX12 Stereo microscope (Olympus, Redmond, WA, USA).

### 2.6. Analysis of hacA Sequence Defects of A. flavus Transformants and Colony PCR of hacA RNAi Transformants

Direct colony PCR without DNA purification was performed using a Phire Plant Direct PCR Master Mix (ThermoFisher Scientific, Waltham, MA, USA). Primary transformants were transferred onto PDA plates using a fine pointed toothpick to lightly touch the conidia-bearing head of a single conidiophore under a dissecting microscope and carefully transfer the conidia. After growing at 30 °C for two to three days, pin-size white, fluffy young mycelia were collected with a sterile toothpick and transferred to a 0.5-mL microfuge tube containing 30 µL of dilution buffer. The contents were disrupted by up-and-down strokes with the toothpick. A 0.5 µL aliquot of the resulting solution was used for PCR amplification in a final volume of 20 µL. The PCR condition consisted of an initial denaturation at 98 °C for 5.0 min, followed by 40 cycles of denaturation at 98 °C for 5 s, annealing at 65 °C for 5 s, and extension at 72 °C for 30 s. PCR products were purified with a DNA Clean & Concentrator-5 kit (Zymo Research, Irvine, CA, USA) and sequenced at the Iowa State University DNA Facility (Ames, IA, USA). For colony PCR of *hacA* RNAi transformants, a primer set, StemF_P and CkRloop (Appendix A), specific to the right-side partial loop and stem portion of the RNAi constructs, was used. PCR fragments were examined by 1.5% agarose electrophoresis in TAE (Tris–acetate–EDTA) buffer.

### 2.7. Semi-Quantitative Thin Layer Chromatography (TLC) Analysis of Aflatoxin

Two agar plugs from each PDA plate containing different transformants were cored using a Transfertube (Spectrum, Houston, TX, USA) and placed in a 2.0 mL microtube. Aflatoxin and other metabolites were extracted overnight with 0.5 mL acetone, followed by the addition of 0.5 mL chloroform. An equal volume of 0.4 mL extraction solution from each microtube was transferred to a new 0.5-mL microtube, and the contents were air dried. TLC was performed on a Silica gel 60 plate (EMD Chemicals, Gibbstown, NJ, USA). The metabolite extracts were first dissolved in 10 µL of acetone and spotted onto the plate; another 10 µL of acetone was added to remove residual extracts. The TLC plate was developed using a toluene–ethyl acetate–acetic acid (60:35:5, *vol*/*vol*/*vol*) solvent system [48], and the image on the UV transilluminator (Fotodyne, Hartland, WI, USA) was recorded using an iPhone 13.

## 3. Results

### 3.1. The Non-Conventional Intron in A. flavus hacA Is 20 Nucleotides Long and Its Removal Yields a Truncated HacA Lacking α-Helical Structures at the Carboxyl Terminal Region

The A. *flavus hacA gene* is 1371 base pairs in length. It encodes a predicted HacA protein that contains a region rich in basic and acidic amino acids (positions 41 to 89; with a ratio of 17:9) as well as a canonical bZip domain (Figure 1A). Analysis of our in-house bioinformatics database, which included RNA-seq reads from *A. flavus* isolates of NRRL3357, CA14, KD17, and KD53, revealed that the full-length *hacA* gene contained a single spliced intron of 54 nucleotides (nucleotides 219 to 272) with the consensus GT…AG splicing sequence. However, many sequence reads were missing 20 nucleotides at a specific location in *hacA* (around nucleotides 701/706 to nucleotides 720/726) flanked by a hexanucleotide direct repeat, CTGCAG, which is a *Pst*I restriction site (data not shown). These shortened reads are likely to have resulted from the removal of the non-conventional intron, a finding previously reported for the *hacA* genes of other fungi. Comparison of the *hacA* non-conventional intron sequences from *Aspergillus parasiticus*, *A. oryzae*, *A. fumigatus Aspergillus nidulans*, and *A. niger* [7,11,49] showed that, like *A. flavus hacA,* their non-conventional introns are also 20 nucleotides long and flanked by two *Pst*I restriction sites (Figure 1B). However, the exact cleavage positions remain ambiguous because the removal of 20 nucleotides between the two *PstI* sites yields the same product. The introns in the respective *hacA* genes of *Trichoderma reesei* and *Verticillium dahliae* are also 20 nucleotides long, while those of *N. crassa*, *U. maydis*, and *S. cerevisiae* are larger and vary in size (data not shown) [11,20,50]. The wild-type HacA protein contains 438 amino acids, while due to the frameshift caused by the removal of the non-conventional intron (Appendix A), the predicted HacA, designated HacA^∆^, is truncated and consists of 345 amino acids. The protein designations for *A. flavus* HacA and *A. oryzae* HacA (=HacA^∆)^ in the UniProKB database are B8NL91 and Q1XGE2, respectively. The 3D models generated by AlphaFold [38,39] (date of access: 28 February 2023) reveal that both HacA and HacA^∆^ contain an α-helical bZip domain (Figure 1C). Additionally, HacA features a region with extensive α-helical structures from reside 245 Thr (T) to residue 410 Leu (L), whereas these α-helical structures are nearly absent in HacA^∆^, leaving only a small residual portion.

### 3.2. CRISPR/Cas9 Mutagenesis of the hacA Functional Domain-Coding Regions and the Non-Conventional Intron Generates Mutants Exhibiting Various Phenotypes

To elucidate the relationships between changes in genotype and the resulting phenotypes, CRISPR/Cas9-based random site mutagenesis experiments were conducted on the basic and acidic amino acid-rich region (Bsc), the bZip domain, and the non-conventional intron sequence (Nt20) (Appendix A). Transformation results indicated that total numbers of primary transformants in the experimental sets compared with the respective control sets were remarkably low (Appendix A and Table 2). Notably, no transformants were generated from the bZip1 set, and only one transformant was obtained from the Bsc3 set. The regeneration ratios of the transformants in the experimental sets, using the total transformant number of respective control sets as a reference, reached only about 16%. Sequencing of randomly selected transformants revealed that deletions and an insertion in the Bsc-coding sequence were all in-frame mutations (Figure 2A). All deletions in the bZip-coding sequence were also in-frame (Figure 2B). In contrast, sequence mutations in the Nt20 sequence were predominantly out-of-frame insertions or deletions, with one notable large deletion of 769 bps and a significant insertion of approximately 5 kb (Figure 2C). Among the *hacA* mutants, four main types of colony morphology were observed based on growth on PDA: (i) the wild-type conidial type (designated as C), producing abundant conidia, including variants that produced small amounts of sclerotia (designated as Cs); (ii) mixed type (designated as CS), producing intermediate levels of conidia and sclerotia; (iii) sclerotial type (designated as S), including variants that produced low levels of conidia after prolonged growth (designated as Sc); and (iv) mycelial type (designated as M), which barely produced conidia and sclerotia (Figure 3).

### 3.3. The hacA Mutants Display Different Sensitivities to DTT

DTT is a potent reducing agent that inhibits disulfide bond formation, leading to the accumulation of unfolded proteins in cells and triggering ER stress [20,22]. On complex PDA and minimal CZ plates supplemented with 10 mM DTT, the growth of wild-type CA14 was notably reduced. In comparison to the DTT controls, the growth of *hacA* mutants, which displayed various levels of conidiation, was somewhat affected by DTT, as evidenced by smaller colony sizes. However, the growth of the mycelial and sclerotial types of mutants was severely reduced on both DTT plates (Figure 3). The rich nutrient content of PDA appeared to exacerbate ER stress in the mycelial and sclerotial mutants from the Bsc and bZip sets, resulting in no growth even after a week of prolonged incubation.

### 3.4. The hacA Mutants All Produced Aspergillic Acid and Kojic Acid but Not Anthraquinones or Aflatoxin

To evaluate how phenotypic and genotypic changes in the *hacA* mutants affected secondary metabolism, the production of aspergillic acid, anthraquinones, and kojic acid were assessed on differential media: ADM, CAM, and KAM, respectively (Figure 4). All *hacA* mutants, irrespective of their phenotypes and genotypes, produced both aspergillic acid and kojic acid. Regarding anthraquinone production, the mycelial mutants from the bZip3 and Nt20 sets produced significantly lower levels compared with both the sclerotial and conidial mutants, as well as the mycelial mutants from the Bsc set, as judged by the fluorescence intensity under long wavelength ultraviolet light. The characteristic yellow-pigmented florescent anthraquinones observed on CAM are believed to be intermediates in the aflatoxin biosynthesis pathway [43]. Consequently, aflatoxin production on PDA by selected *hacA* mutants was assessed using semi-quantitative TLC analysis (Figure 5). The conidial and sclerotial mutants produced amounts of aflatoxins B_1_ and B_2_ comparable to those of wild-type CA14. In contrast, the mycelial mutants, except for one mutant (bZip∆42) that produced a reduced amount of aflatoxin B_1_ but no detectable B_2_, did not produce discernible amounts of aflatoxin B_1_ or B_2_ under the same analytical conditions. Further examination of this unique mutant through serial transfers on PDA indicated that it probably contained heterogeneous nuclei, as indicated by sectoring of a fast-growing conidial area on the fourth transfer (Figure 5B). A fifth transfer of a small piece of clean mycelial agar onto a PDA plate restored the original mycelial morphology; subsequent TLC analysis confirmed that it did not produce aflatoxins.

### 3.5. The hacA RNAi Affects Regeneration and Development of A. flavus Protoplasts

To investigate the impact of *hacA* RNAi on *A. flavus* CA14 survival, two versions of RNAi constructs (Appendix A) were transformed into varying amounts of protoplasts, to evaluate their effects on protoplast regeneration and development (Appendix A). Table 3 shows that the averaged regeneration rates using the first *hacA* RNAi construct ranged from 45 to 65% of the controls. Most *hacA* RNAi transformants when transferred from CZ plates to PDA plates exhibited colony morphology similar to that of the control sets, which was fairly uniform. However, a small number of the *hacA* RNAi transformants displayed intense green, velvet-like colony morphology (Appendix A). PCR results indicated that the *hacA* RNAi cassette was integrated into the genomes of the *hacA* RNAi transformants (Appendix A). Total transformants generated from the second *hacA* RNAi construct (*hacA*i) were accurately enumerated. The overall survival rates of the transformed protoplasts ranged from 71 to 93% (Table 4). Control transformants maintained uniform colony morphology on PDA plates. In contrast, among the *hacAi* transformants, in addition to the previously observed intense green, velvet-like morphology, a few exhibited more sclerotial forms with reduced conidiation, while one transformant displayed retarded growth (Figure 6).

## 4. Discussion

Small mutations generated by CRISPR/Cas9 gene editing demonstrate that subtle changes in *A. flavus hacA* can significantly affect cellular growth, conidiation, and sclerotial production. These changes reveal genotype–phenotype relationships that are not easily achieved through convention gene deletion and disruption techniques. The low numbers or absence of transformants regenerated in the Bsc and bZip experimental sets suggest that all out-of-frame sequence defects in these coding regions are lethal to the cells. Previous research has confirmed the involvement of *Aspergillus* HacA in cell wall integrity [23,51]. In *T. rubrum*, a marked impairment in protoplast regeneration was observed for the Δ*hacA* mutant [19]. Thus, the detrimental effects of out-of-frame mutations are likely to result from the formation of defective HacA, which lacks one or both functional domains and disrupts vital cellular processes. Conversely, many of the surviving in-frame mutants probably produce HacA with altered protein conformation, impacting growth and developmental processes, as indicated by mutants with various levels of conidiation and sclerotial production. The integrity of the intron sequence is also critical for *A. flavus* survival and development. For example, Nt20 mutants with single nucleotide indels within the intron displayed varied conidiation; one deletion mutant was sclerotial (Figure 2C). These results suggest that subtle changes can hinder correct intron processing. Otherwise, the removal of introns containing these indel mutations would still yield normal, mature *hacA* transcripts. In yeast, a transcriptional switch is necessary to release *hacA* mRNA from a translational block controlled by pairing of the unconventional intron with its 5′ untranslated region upon ER stress [52]. In *A. niger*, splicing of its non-conventional intron is associated with truncation of 230 nucleotides at the 5′-end of the *hacA* transcript [49]. It remains unclear whether defects in the *A. flavus hacA* intron sequence can impact this mechanism, if it exists.

Upon close examination of phenotypes and genotypes of the *hacA* mutants, a few generalizations can be made as follows: (i) the conidial type is primarily associated with defects in the Bsc-coding and intron sequences, while the mixed and sclerotial types are mainly linked to defects in the bZip-coding sequence; (ii) sequence defects shifting from the Bsc-coding region to the bZip-coding region in the mutants resulted in a transition from conidial to more sclerotial phenotypes; and (iii) the mycelial type, regardless of mutation location, was primarily associated with large deletions or insertions. A previous study on *A. flavus hacA* demonstrated that only knockouts containing heterogeneous nuclei (i.e., the wild-type and ∆*hacA*) were viable [24]. In a subsequent study, a ∆*hacA* mutant lacking the entire coding region was generated from *A. flavus* NRRL3357 [23]. This mutant exhibited severely retarded mycelial growth and was unable to produce conidia and sclerotia, resembling the mycelial mutants described in this study. Similarly, deletion of the entire and most of *hacA* in *A. oryzae* [22] and in *A. niger* [53], respectively, generated mutants that were severely retarded in growth and defective in conidiation. In this study, the pleotropic effect also arose from small mutations in the Bsc- and bZIP-coding regions, as well as large deletion and insertion around the non-conventional intron. Changes in the functional domains similarly resulted in sclerotial mutants, highlighting the significance of HacA conformation in regulating vital cellular and developmental processes. The response to DTT-induced ER stress varied among the aforementioned mutant types. On complex PDA and minimal CZ plates supplemented with DTT, the growth of the conidial mutants was somewhat affected, while the growth of the mycelial and sclerotial mutants was severely impaired. Notably, the sclerotial mutants from the Bsc and bZip sets showed no growth even after a week of prolonged incubation. Sclerotia, resting structures formed by the aggregation of hyphae into dense, pigmented propagules, remain viable for extended periods under adverse environmental conditions [54]. It is surprising that these sclerotial mutants were more sensitive to DTT than the conidial mutants—a phenomenon not previously observed.

Fungal development and secondary metabolism are interconnected through shared G-protein signaling pathways and other common regulators [55,56]. Research on *hacA/hac1* has primarily focused on growth and development aspects, including virulence and pathogenicity, with less emphasis on secondary metabolism. Recently, Yu et al. [40] concluded that *A. flavus* ∆*hacA* mutants were unable to produce aflatoxin, consistent with the mycelial mutants in this study that also did not produce aflatoxin. All mycelial mutants, like the conidial and sclerotial mutants, produced aspergillic acid and kojic acid, but some exhibited variations in the production of anthraquinones believed to be precursors of aflatoxin [43]. The reason that the Bsc mutants still produced anthraquinones is unclear. The bZip domain of HacA binds to a consensus UPR element in the promoters of ER-stress responsive genes, activating gene expression [5]. Genome-wide gene expression studies have shown that HacA also coordinates many vital cellular processes and responses [22,23,57,58]. However, how HacA, via the UPR pathway, controls the cascade regulation of secondary metabolite production remains to be understood. The *A. flavus* NRRL3357 ∆*hacA* mutant lacks HacA due to the deletion of the entire *hacA* gene, whereas the CA14 mycelial mutants probably produce altered conforms of HacA, either with or without an intact bZip domain, which is also true for the sclerotial mutants. Therefore, impairments in growth, development, and the aflatoxin biosynthesis pathway may be pleiotropic effects specifically attributed to the loss or alteration of HacA function. In other words, HacA may not significantly influence overall secondary metabolism.

Despite limited but promising results in targeting genes in the aflatoxin biosynthesis pathway to reduce preharvest aflatoxin contamination using HIGS [31,32,33,35], a key critique on this aflatoxin-focused RNAi strategy is that infected grains may become moldy or produce a musty odor under poor storage conditions if fungal growth is not adequately suppressed [59]. Studies on transgenic maize targeting *A. flavus* genes unrelated to the aflatoxin pathway such as α-amylase gene, alkaline protease gene, or polygalacturonase gene have shown that RNAi targeting these genes reduces both fungal colonization and aflatoxin accumulation [60,61,62]. Moreover, expression of a tachyplesin-derived synthetic peptide to control *A. flavus* growth and aflatoxin production resulted in up to a 72% reduction in fungal growth in transgenic seeds compared with isogenic negative control seeds [63]. Practically speaking, RNAi is a knock-down and not a knockout technique [64,65], and its effects are temporary [66]. The integration locations in the genome and temporal expression of the vectors certainly affect the levels of miRNA production. In this study, RNAi experiments in *A. flavus* revealed a general trend of decreased numbers of transformants regenerated from protoplasts. This trend may partly have been due to the degradation of *hacA* transcripts by yet-to-be-identified *hacA* miRNAs. The partially degraded *hacA* transcripts are likely to have yielded truncated HacA proteins, similar to those produced in the non-surviving transformants from the CRISPR experiments, which presumably contained *hacA* genes with various out-of-frame mutations. We can anticipate the adverse effects of these conformation-changed, function-altered HacA proteins on the UPR pathway. Regardless of whether the sequence changes occur at the DNA or transcript level, the outcome remains the same; the accumulation of unfolded or misfolded proteins would severely disrupt ER protein homeostasis, leading to toxic and lethal consequences for the cells. The *hacA* mutants of *T. rubrum* and *A. flavus* have also been shown to be unable to colonize corn kernels [19,23], suggesting the potential of targeting *hacA* to reduce fungal loads in infected crops. Therefore, a holistic RNAi approach including designing a multiplex construct targeting genes critical to fungal growth, development (such as conidiation and sclerotial formation), and aflatoxin biosynthesis would be a practical and effective strategy.

## Figures and Tables

**Figure 1 jof-10-00719-f001:**
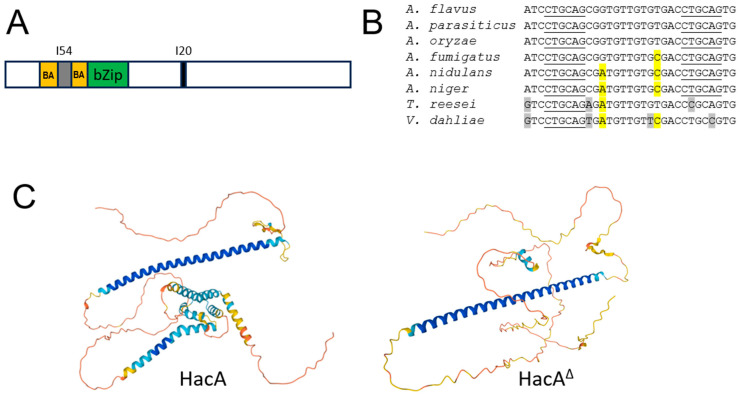
The *A. flavus hacA* gene. (**A**) Schematic representation of the functional domain-coding regions and introns. Gray and black areas indicate the single conventional intron (I54) and the non-conventional intron (I20 = Nt20). BA indicates the coding region rich in basic and acidic amino acids, which is interrupted by the conventional intron. bZip denotes the basic leucine zipper coding region. (**B**) Comparison of the 20-nucleotide non-conventional introns of *A. flavus*, *A. parasiticus*, *A. oryzae*, *A. fumigatus*, *A. nidulans*, *A. niger*, *T. reesei*, and *V. dahlia*. The hexanucleotide repeat, which is the *Pst*I restriction site sequence, is underlined. Single nucleotide polymorphisms are highlighted. (**C**) The 3D models of HacA and HacA^∆^. The amino acid sequence from residues 83 to 146 is the bZip domain, which is the long α-helical structure in both forms. AlphaFold produced a colored confidence score for each residue in the α-helical structure. The score range of each color is blue > 90, light blue > 70 but <90, yellow > 50 but <70, orange < 50. Refer to Appendix A for both amino acid sequences.

**Figure 2 jof-10-00719-f002:**
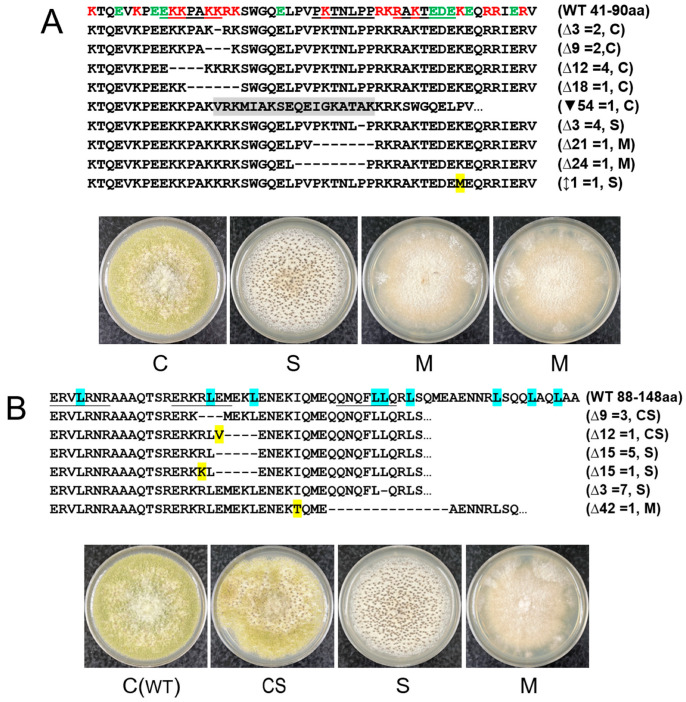
Phenotypic changes resulting from sequence mutations in the *hacA* functional domain-coding regions and the nonconventional intron. (**A**) The basic/acidic domain: K (lysine) and R (arginine) are basic amino acids, while E (glutamic acid) and D (aspartic acid) are acidic amino acids. The three target regions are underlined. (**B**) The bZip domain: The domain forms an α-helix (see Figure 2) with leucine (L) residues highlighted in blue. The three target regions are underlined. The wild-type (WT) CA14 strain is shown alongside the CS mutant for visual comparison and consistency in presentation. (**C**) The non-conventional intron: Both amino acid sequences of the wild-type CA14 strain before and after the removal of the 20-nucleotide intron are shown. Only the predicted amino acid sequences before the removal of the respective introns from the mutants with single-nucleotide insertion or deletion are shown; the removal of these introns yielded amino acid sequences identical to that of the truncated HacA^∆^. Single amino acid changes are highlighted in yellow. Missing amino acids are indicated by dashes, and inserted amino acids are highlighted in grey. A star indicates the end of an amino acid sequence. In the three panels, open triangles and solid inverted triangles followed by numbers indicate deleted and inserted nucleotides, respectively. The double-headed arrow indicates nucleotide substitution. The numbers after the equals signs are the number of sequenced transformants with the same mutation. The following designations are used for colony morphology: C, conidial; S, sclerotial; CS, a roughly equal mix of conidial and sclerotial, Cs, more conidial than sclerotia, Sc, more sclerotial than conidial, and M, mycelial. Mycelial mats barely contained conidiophores. The powdery-looking sectors in panels (**A**–**C**) when examined under a dissecting microscope, were found to contain aggregations of hyphae, which appeared to be white sclerotia initials.

**Figure 3 jof-10-00719-f003:**
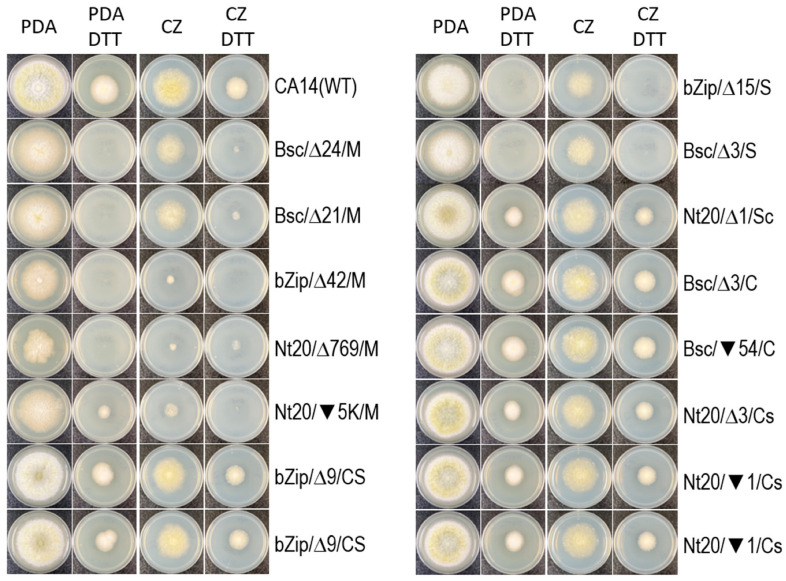
DTT induced ER stress on the growth and development of the wild-type CA14 strain and the *hacA* mutants. Open triangles and solid inverted triangles followed by numbers indicate deleted and inserted nucleotides, respectively. Bsc, bZip, and Nt20 have been added to strain designations in reference to the mutated regions. For details regarding the morphologies (C, CS, S, Cs, Sc, and M) of the Bsc, bZip, Nt20 mutants, refer to the legend for Figure 3. Conidia from wild-type CA14 and various types of the *hacA* mutants were inoculated onto PDA and CZ plates, both with and without the addition of 10 mM DTT. Plates were incubated at 30 °C for four days in the dark.

**Figure 4 jof-10-00719-f004:**
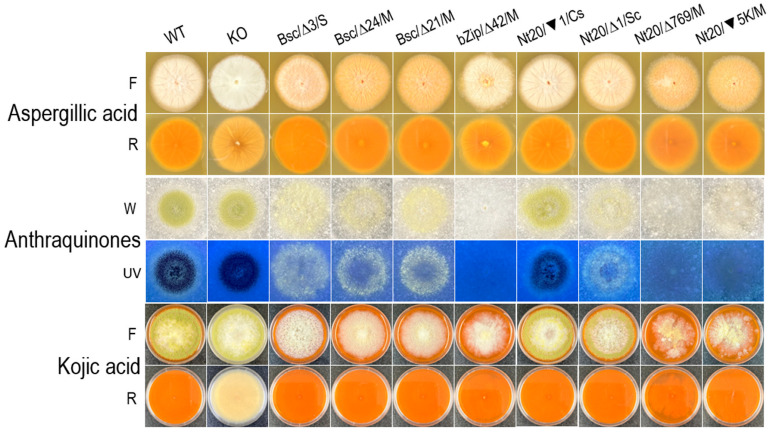
Production of aspergillic acid, anthraquinones, and kojic acid by the *hacA* mutants. Colony morphologies and pigmentation were examined on ADM, CAM, and KAM plates. F, top view; R, reverse side; W, white light; UV, longwave ultraviolet light. Pigmentation in ADM was restricted to the edge of the colony, while pigmentation on KAM (complex formed by diffusible kojic acid with ferric ion) was distributed throughout the plate. WT, wild-type CA14; KO, respective gene knockout mutants. Open triangles and solid inverted triangles followed by numbers indicate deleted and inserted nucleotides, respectively. Bsc, bZip, and Nt20 added to strain designations for reference to the mutated regions. Refer to the legend for Figure 3 for strain details and designations of morphologies.

**Figure 5 jof-10-00719-f005:**
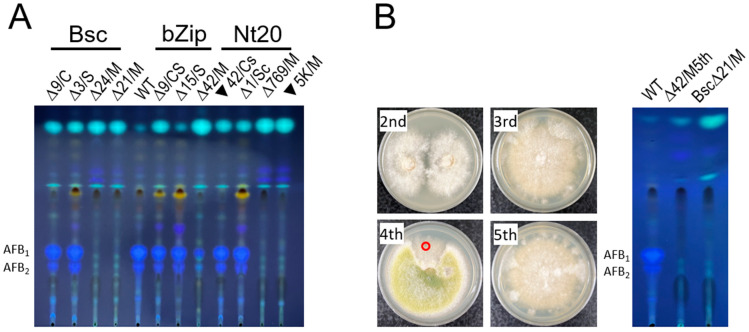
Semi-quantitative TLC analysis of aflatoxin production by selected *hacA* mutants with varying colony morphologies. (**A**) The designations Bsc, bZip, and Nt20 refer to mutants with defects in their HacA functional domains and the non-conventional intron. WT denotes the wild-type CA14 strain. Open triangles and solid inverted triangles followed by numbers indicate deleted and inserted nucleotides, respectively. Refer to the legend for Figure 3 for strain details and designations of morphologies. (**B**) Lack of aflatoxin production by the fifth transfered *hacA* mutant of bZip∆42/M. The mutant was serially transferred onto PDA. A small piece of mycelia-containing agar from the fourth culture (circled red) was transferred onto a PDA plate. Two agar plugs from the fifth culture were cored and analyzed using semi-quantitative TLC. Bsc∆21/M, a mycelial mutant that does not produce aflatoxin, served as a negative control.

**Figure 6 jof-10-00719-f006:**
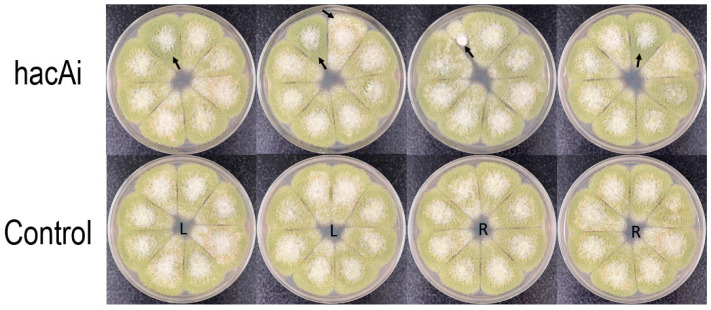
Colony morphology of *hacA*i transformants on PDA plates. Transformants exhibiting sclerotial, green and velvet-like, or retarded growth morphologies are indicated by arrows. L and R denote transformants from the L_bZip and R_bZip control sets. Cultures were incubated at 30 °C for five days in the dark.

**Table 1 jof-10-00719-t001:** Universal primers and protospacers for constructing CRISPR/Cas9 vectors expressing sgRNA cassettes.

Primer	Sequence (5′→3′)	
U6-F-P ^a^	ATACTGCAGTTCTCTTTAGAATTCAACTGTGGGT	
U6-R-K ^a^	TATGGTACCACATATTTAAAAAAAGTCTCCTGCC	
gdFd_gene	Fd_seqGTTTTAGAGCTAGAAATAGCAAGTTAA	
gdRC_gene	RC_seqACTTGTTCTTCTTTACAATGATTTATATACC	
Target	Protospacer sequence ^b^	
	Fd_seq	RC_seq
Bsc1	AGAAGAAGCCTGCAAAGAAG	CTTCTTTGCAGGCTTCTTCT
Bsc2	CCAAGACCAACTTGCCTCCG	CGGAGGCAAGTTGGTCTTGG
Bsc3	TGCCAAAACAGAAGATGAGA	TCTCATCTTCTGTTTTGGCA
bZip1	ACGCGTCCTTCGAAATCGTG	CACGATTTCGAAGGACGCGT
bZip2	GGAGCGCAAAAGGCTGGAAA	TTTCCAGCCTTTTGCGCTCC
bZip3	AGAATCAATTCCTCCTTCAG	CTGAAGGAGGAATTGATTCT
Nt20	AGGTCACACAACACCGCTGC	GCAGCGGTGTTGTGTGACCT

^a^: U6-F-P and U6-R-K were derived from the *A. flavus* U6 promoter and terminator sequences with tagged restriction sites of *Pst*I (CTGCAG) and *Kpn*I (GGTACC), respectively. ^b^: Fd_seq, forward sequence; RC_seq, reverse complement sequence.

**Table 2 jof-10-00719-t002:** Total numbers of regenerated transformants after the *hacA* functional domain-coding sequences were edited by CRISPR/Cas9.

CRISPR	Domain	AA Position ^a^	Transformant #	Sequenced/Indel Type ^c^
Controls			816 ^b^, 850, 900	-
HacA	Bsc1	49–55	48	10/10 in-frame
	Bsc2	66–72	6	6/6 in-frame
	Bsc3	75–81	1	1/single base substitution
	bZip1	88–94	0	-
	bZip2	102–108	55	11/10 in-frame, 1 wild-type
	bZip3	121–127	132	8/8 in-frame
	Nt20	217–223	59	17/16 in-frame and out-of-frame, 1 wild-type

^a^: see Appendix A for amino acid sequences. ^b^: The total of 816 is from Chang [45]. The total of 900 is for the Bsc and Nt20 sets, and the total of 850 is for bZip1 and bZip2 sets (refer to Appendix A for details). Approximately 2.0 × 10^6^ protoplasts and 250 ng of a CRISPR/Cas9 vector were used in each transformation. ^c^: see Figure 3 for additional information.

**Table 3 jof-10-00719-t003:** Estimated total transformant numbers from the initial RNAi experiments using different amounts of protoplasts.

Linearized DNA ^a^	Protoplast Amount	Total Transformant Number ^b^
P70-U6	2.0 × 10^6^	>1000
R_bZip		>1000
RNAi 1		712
RNAi 2		575
R_bZip	1.0 × 10^6^	611
RNAi 1		351
RNAi 2		190

^a^. Approximately 1.0 µg *Hind*III linearized plasmid DNA was used in each transformation. P70-U6 was the cloning vector containing a U6 expression cassette. R_bZip was the construct containing a half-stem–loop structure. Two independent *hacA* RNAi constructs were tested. ^b^. Primary transformants of *A. flavus* CA14 on CZ regeneration plates were visually estimated after cultures were incubated at 30 °C for four days in the dark. Each total number is the sum of transformant numbers from six CZ plates (refer to Appendix A).

**Table 4 jof-10-00719-t004:** Actual total transformant numbers from the *hacA* RNAi experiments using different amounts of protoplasts.

Linearized DNA ^a^	Protoplast Amount	Transformant Number	Survival Rate (%)
L_bZip	~3.0 × 10^4^	590/740 ^b^	75
R_bZip		519/700	
hacAi 1		334/463	
hacAi 2		449/612	
R_bZip	~1.0 × 10^4 c^	104/134	71
hacAi 1		59/75	
hacAi 2		85/116	
L_bZip	~1.0 × 10^4 d^	61/81	93
hacAi 1		61/85	
hacAi 2		51/66	

^a^. Approximately 0.5 µg *Hind*III linearized plasmid DNA was used in each transformation. L_bZip and R_bZip were controls, each containing a half (left or right) of the hacAi stem–loop structure. Two independent mini-prep *hacA* RNAi constructs were tested. ^b^. Primary transformants on CZ regeneration plates were examined and accurately counted under a dissecting microscope. The first count was performed after plates were incubated at 30 °C in the dark for three days, when various sizes and shapes of young colonies began to appear. The counted plates were then placed at room temperature for an additional day before final total transformant numbers were recorded. Each number is the sum of transformant numbers from six CZ plates, ^c^ and ^d^, two independent experiments.

## Data Availability

Raw data and materials described in this study will be shared upon reasonable requests, in accordance with USDA policies and procedures.

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
