# Peer review of "The Aspergillus flavus hacA Gene in the Unfolded Protein Response Pathway Is a Candidate Target for Host-Induced Gene Silencing"

_jof, 2024, doi:10.3390/jof10100719_

Round 1

Reviewer 1 Report

The abstract is impossible to follow and would have to be totally rewritten.  What does it mean that the changes were “exclusively in frame”? – this is only understandable in the context of the paper.  In this case the issue of the missing/present 20 base pair sequence is relevant, but this has not been presented to the reader. And immediately noting transformant frequencies being less than controls has absolutely no context.  An abstract has to stand on its own; this one is incomprehensible without having first read the paper, and even then it is hard to follow.

There are several issues with the style.  I have just noted a very few  

There are lots and lots of missing definite articles.  Just in paragraph 1 of the intro we have

Line 31 The endoplasmic … ; line 34  The UPR… ; line 37 The UPR… ; line 38 Among them the IRE… ; line 41 Disruption of the yeast …  The paper will have to be checked over completely by a native English speaker to correct this issue.

Lots of other style mistakes 

Line 39 should be “Central to this … , which has been …

Line 46 In U. maydis it has been

Line 48  As with genes of yeast and mammalian cells, those of filamentous fungal…

Line 53 maybe a missing reference for (2022)

As noted, the paper needs careful overall editing for sentence structure.  Sometimes the issues obscure meaning, but generally they just influence the smoothness of reading.

Author Response

Reviewer 1

I thank the reviewer for the valuable comments.

Introduction

There are a number of technical complexities that are not covered in the introduction and in the flow of the results that make interpretations problematic. Without this context many questions arise that are either not addressed or addressed much later in the paper.

Response: I am unclear about what “technical complexities” refer to. Since reviewer later mentioned donor DNA, I assume this pertains to CRISPR/Cas9. I have added a brief explanation of this technique, including the role of donor DNA in specific applications, to the introduction (lines 91-100).

For example, the critical existence of null mutants of hacA should be presented early, potentially in the introduction, but certainly at the start of the section on mutant isolation.

Response: As suggested, I have included text about null mutants of hacA in yeast and fungi (lines 61-71).

This result makes the observation that the modifications in the acid/basic and bZIP regions lead only to in-frame deletions or insertions interesting, because there seems to be no reason that totally inactive, frame-shifted proteins would fail to show up during the mutagenesis if the protein is not essential. It could be that the partial out-of-frame protein messes with cell regeneration during transformation – this idea can be teased from the manuscript with difficulty.

Response: I agree that his is a plausible explanation for the low survival rates of the regenerated transformants. I consider this as well, but proving it is challenging.

Sometimes non-functional, or partially functional proteins can act as “dominant negatives” with the mutant protein interfering with an otherwise functional system. It appears A. flavus is polynucleate. How does this behavior complicate things – can you have a cell with both mutant and WT genomes?

Response: While A. flavus is polynucleate, the transformants generated are predominantly mononucleate. Protoplasts with multiple nuclei are uncommon; most contain only one nucleus, and some may have none. Otherwise, creating a gene disruption mutant will be challenging due to the contamination by the wild-type nucleus.

A statement in the discussion notes that hacA deletion mutants have to have heterogenous nuclei –

Response: The result regarding “heterogeneous nuclei” is indeed incomplete. It likely arises from a combination of low transformation frequency and rare protoplast fusion events.

surely this is an important point, and a discussion of this point belongs in the introduction.

Response: A subsequent study on the generation of an A. flavus hacA deletion mutant, along with known hacA mutants from other fungi, is mentioned in the introduction (lines 62-71)

We are told the A. flavus sequence without the 20 bp non-conventional intron insert looks just like the A. oryzae HacA, so this insert/deletion is different among the Aspergilli? This seems to vary – this point has to be made clearly in the introduction.

Response: The hacA nucleotide sequences of A. flavus and A. oryzae are100% identical. However, the annotated proteins in the UniProKB database are based on different mRNA transcripts: the wild-type transcript for A. flavus HacA and the intron processed (removed) transcript for A. oryzae HacA. This allows us to use A. oryzae HacA as a proxy for the truncated A. flavus HacA. Aspergillus species commonly have a non-conventional intron of 20 nucleotides

The starting data set shows that RNA transcripts from A. flavus hacA have the standard 54 base pair intron missing relative to the DNA sequence, but many transcripts were also missing a 20 bp sequence in the DNA that is flanked by PstI sites, such that recombination between the PstI sites would loop out this sequence, leaving behind a single PstI site. This sequence is defined as a non-conventional intron, implying the loss of information seen in the transcripts relative to the defined genome sequence is the result of an event occurring at the RNA level, rather than at the DNA level, and we are told that this unconventional intron is found in other Aspergillus species, and not processed by the conventional processing mechanism. It would be very useful to have a section in the introduction defining how it is known that the event is occurring at the RNA level, and what, if anything, is known about the unconventional processing mechanism.

Response: Please refer to the added Figure 1A for the location of the non-conventional intron in the A. flavus hacA gene. Conventional introns, processed by the spliceosomal mechanism, are characterized by the consensus splicing sequence "GU" at the 5' splice site and "AG" at the 3' splice site. In contrast, the non-conventional intron is cleaved by a kinase-nuclease factor (the IRE1 protein, lines 52 and 58). The text has been modified to clarify the processing of the non-conventional intron (lines 51-53).

How do we know that the sequence is not an unconventional “transposon” that hops in and out at the DNA level?

Response: This is an interesting proposition. The smallest known transposons in fungi are several hundred base-pairs long. Thus, the size of the intron suggests otherwise.

Results 

Figure 1A is said to show the unconventional intron but the figure is impossible for me to follow. Panel A has the amino acids with some defined by nucleotides, others not; perhaps the section where the amino acids are defined by the nucleotides is the intron?

Response: It’s a snipped image from the JBrowse genome viewer, showing both strands of the intron region and amino acid sequences translated from three reading frames. The original Figure 1A, a redundant presentation of Figure 1B, has been deleted and replaced by a diagram showing the entire hacA gene structure.

The “intron” is said to be defined by a black line, which I guess are the bars in the transcript reads shown in the second panel of 1A. What are the lines in blue?

Response: Yes. In paired-read sequencing, each DNA fragment is sequenced from both ends. Pink and blue lines indicate reads sequenced from different ends.

I would guess about 50% of the reads lack the 20 nucleotide intron. Are the two forms found in the same cell, or are there some cells with the intron and some without? Is the larger insertion in S. cerevisiae also flanked by a repeat sequence? Given that this intron identification is important to the paper this defining figure needs to be much better laid out.

Response: In the study by Cox and Walter (Cell, Vol. 87, 391–404, 1996), both forms (wild-type and processed) were present after induction at a ratio of about 1:3. Therefore, not all cells behave the same, and growth dynamics may also play a role. Figure 1B shows that the introns in the T. reesei and V. dahliae hacA genes are not flanked by a repeat sequence, unlike the Aspergillus introns. Thus, the presence of two repeat sequence flanking the Aspergillus hacA intron likely a coincidence, not a must for correct intron processing.

The next section defines the potential 3D structure of the proteins with and without the 20 bp intron. All the sequence before the start of the intron is the same in both proteins, and they get very similar structures, dominated by an alpha-helix that represents the DNA binding bZIP domain. The longer, intron-containing sequence shows considerably more structure (several alpha helices) than does the smaller, frame-shifted construct arising from the intron removal. Alpha fold makes good predictions, but currently does not encompass the structural implications of interactions with other proteins, so overall this information may not be that useful.

Response: The predicted 3-D structures of the wild-type and truncated HacA are provided to for those interested in protein conformation studies. The original Figure 2 is now Figure 1C.

The next effort is to use CRISPR-Cas9 processing to manipulate the structure of the gene defined by the intron, by the bZIP domain, and by the acidic/basic region of the protein, and investigate the phenotypic consequences of these manipulations. The planned modifications are provided as supplementary figures, and these are not particularly clear. Are the residues shown in yellow deleted, modified or just highlighted? Bsc1, Bsc2/3 are the oligos defining the manipulations?

Response: The yellow-highlighted amino acids indicate modified residues compared to the wild-type sequence (see the top sequences in Figure 2). The underlined sequences are target sites in the hacA gene, corresponding to the protospacer sequences shown in Table 1. Please refer to Figure S3B-D for both DNA and amino acid sequences.

Table 2 defines the mutations – these are deletions? The term in frame in the sequenced/indel type column I is not obvious. This information becomes clear with the next part of the figure, but is initially confusing.

Response: For all CRISPR work, sequencing is required to determine actual mutations (insertions and deletions). Table 2 summarizes the mutations concisely and should be read alongside Figure 2.

In many fungal systems where CRISPR is employed (yeast, C. albicans, S. pombe, K. marxianus …) the manipulation uses a targeted Cas9 to cut, and a recombination oligo to direct the exact new sequence desired. Here (not so easy to sort out) I understand the system targets Cas9 to the desired site, and sequences are identified in the surviving colonies that have created changes in the targeted region. This just needs to be clearly laid out in the introduction to section 3.3 by defining the term “random site mutagenesis” in a bit of detail.

Response: The CRISPER technique, developed for gene-editing about two decades ago, has been refined for various biological systems. The “naked” CRISPR (nuclease plus RNA) creates small random mutations at/around a target site, disrupting the gene of interest. In specific applications, donor DNA is included to create predefined (non-random) sequence mutations.

Another critical point that is brought up later, but as already noted needs to be placed right at the beginning of the section on the analysis of mutants, is that a KO strain has been constructed and its phenotype analyzed.

Response: To accommodate the reviewer’s comment, I have revised the subtitle of section 3.2- CRISPR/Cas9 mutagenesis of the hacA functional domain-coding regions and the non-conventional intron generates mutants exhibiting various phenotypes.

The analysis of the CRISPR generated mutants found it interesting that the frequency of transformants after mutagenesis was low. It would be important here to know if the construction of the KO was similarly coupled to low frequencies of identified transformants. How was the KO constructed, how were they identified, what is the actual structure of the KO?

Response: I assure the reviewer that all CRISPR constructs, and the transformation and selection protocols are fine. This CRISPR/Cas9 methodology for A. flavus gene function is well established and has been used not only to create single-gene knockouts but also to generate triple- and quadruple-gene knockouts. The only difference among all CRSIPR constructs used in this study and those used by other researchers is the “protospacer sequence” (~20 nucleotides). Please refer to the sequences listed in Table 1. The lab’s transformation and selection protocols have been in use for decades. I do agree that the low survival rates cannot be fully explained at this moment.

A subsequent observation is that the survival rates of the transformed cells are low, with some targeted regions not generating any viable transformants. This could be because lots of cuts are not repaired leading to death, or that the repair still leads to a lethal structure. We can infer that it is not simple lack of function that is causing the issue, because the null is viable.

As stated earlier, the likely reason for the low survival rates is the formation of altered HacA, which disrupts vital yet-to-be-identified cellular processes. In the discussion, the statement that “…In T. rubrum, a marked impairment in protoplast regeneration was noticed for the ΔhacA mutant (Bitencourt et al., 2020)….“ provides some explanation for this phenomenon, even though mutants lacking HacA are viable (but severely growth-retarded). Since all experiments included control sets (please see Figure S4), which contained hundreds of regenerated transformants, it suggests that mutations in the HacA functional domain-coding regions and the non-conventional intron are primarily lethal. Cells should be able to repair these DNA damages, as indicated by the control sets. It is probable that even if sequence lesions are repaired, they remain lethal, possibly due to interference at the RNA or protein level, or both.  

The other observation is that essentially all the modifications hitting the bZIP domain or the acid/basic regions lead to in-frame deletions (one in-frame insertion). The mutations in the nonconventional intron did not seem to be constrained for frame. Depending on the actual event, the colony phenotype can be different. The manuscript suggests classes of morphology (conidial, sclerotial, different ratios of these two, and mycelial. The colony growth shows that the consequences of the mutations. Why in Figure 3 are we not told that WT gives the conidial phenotype until panel B?

Response: The CA14 morphology was first mentioned in the M&M section: “Wild-type A. flavus CA14 (SRRC 1708), an L-morphotype isolate known for producing abundant spores, along with derived hacA mutant strains....” (lines 111-112). To maintain consistency, the wild-type photo was arranged in panel B to match the layout of panels A, and C. The legend of Figure 2B has been updated to include: “The wild-type (WT) CA14 strain is shown alongside the CS mutant for visual comparison and consistency in presentation.”

What is the phenotype of the deletion mutant? Mutations of the basic/acidic region seem less impactful than those of the bZIP domain, while the mutations in the unconventional intron all generated a non-WT phenotype.

Response: Most of the Nt20 surviving mutants maintain relatively normal conidiation capacity (i.e., Cs rather than CS). Only one mutant with a single nucleotide mutation is sclerotial, but two types of sclerotial mutants (S and CS) were generated in the bZip set (Figure 2B). Notably, the Nt20 mycelial mutants result from a large deletion and a large insertion. These results indicate that conidiation and sclerotial formation are separative pathways controlled by yet-to-be-identified factors.

The mutants were also assessed for response to DTT and for production of aspergillic acid, anthraquinones and kojic acid, and aflatoxin. Aflatoxin is not made in the mycelial mutants.

Response: Yes. DDT, a reducing agent, can break disulfide bonds in proteins, causing cellular stress. Aflatoxin (anthraquinones), kojic acid, and aspergillic acid are classified as secondary metabolites. Over the years, aflatoxin has been used as a model for studies of secondary metabolism. The work demonstrates that the URP regulates the production of secondary metabolites differently.

Next the author investigates the potential of RNAi in manipulating cell function, since the idea is that such a treatment could be used to manipulate A. flavus and serve as a crop protection agent. This is the actual topic of the special issue the paper is directed to. The RNAi directed at hacA do seem to generate phenotypic differences in the transformants.

Response: Yes. The ultimate goal of A. flavus gene function studies over the past three decades has been to develop practical approaches to control aflatoxin contamination in crops, which significantly impacts global economy and health of humans and animals.

Major comments

The abstract is impossible to follow and would have to be totally rewritten.  What does it mean that the changes were “exclusively in frame”? – this is only understandable in the context of the paper.  In this case the issue of the missing/present 20 base pair sequence is relevant, but this has not been presented to the reader. And immediately noting transformant frequencies being less than controls has absolutely no context.  An abstract has to stand on its own; this one is incomprehensible without having first read the paper, and even then it is hard to follow.

Response: The abstract has been rewritten for improved clarity and flow. I hope that the revised version meets the reviewer’s expectations. The pharse “exclusive in frame” has been changed to “all observed mutations in both coding sequences were in-frame.” (line 16)

Detail comments

There are several issues with the style.  I have just noted a very few  

There are lots and lots of missing definite articles.  Just in paragraph 1 of the intro we have

Line 31 The endoplasmic … ; line 34  The UPR… ; line 37 The UPR… ; line 38 Among them the IRE… ; line 41 Disruption of the yeast …  The paper will have to be checked over completely by a native English speaker to correct this issue.

Lots of other style mistakes 

Line 39 should be “Central to this … , which has been …

Line 46 In U. maydis it has been

Line 48  As with genes of yeast and mammalian cells, those of filamentous fungal…

Line 53 maybe a missing reference for (2022)

As noted, the paper needs careful overall editing for sentence structure.  Sometimes the issues obscure meaning, but generally they just influence the smoothness of reading.

Response: The manuscript has been extensively revised and edited. I hope that it now represents a significantly improved version.

Reviewer 2 Report

In this contribution, the authors aim to explore the function of the HacA protein in Aspergillus flavus by targeting three specific regions of its coding sequence (a basic amino acid/acid-enriched domain, the bZIP domain and a 20 bp intron) using CRISPR/Cas9 and RNAi gene silencing. The study reveals clear phenotypic differences, both morphological (colony types) and metabolic, particularly in the production of secondary metabolites of medical importance, such as aflatoxins. Hence, A. flavus hacA may be considered a good candidate for future host-induced gene silencing to control aflatoxin infection and contamination of crops and endanger human and animal health. However, before being accepted for publication in the Journal of Fungi it should address some opportunities for improvement to make the article more understandable.

Writing. While it is possible to follow the main ideas of the results, there are sections that are challenging to interpret for several reasons (including unfinished sentences, missing references, typographical errors, ambiguity of ideas, and lack of narrative flow). Therefore, is would strongly recommend a thorough round of proofreading and editing to improve clarity.

Figures. There should be a consistent and clear notation for all mutants across all figures and experiments. All the details of this notation should be clearly explained in the figure captions. Additionally, images of genomic views (such as read mappings) should show the genomic context (location and gene span) and be presented at a high resolution for clarity.

Author Response

Reviewer 2

I thank the reviewer for the valuable comments.

Title

Although the study proves that the targeted regions contribute to different phenotypic outcomes, it is suggested modifying the manuscript title. The results do not establish a single phenotype -or function- that can be attributed to each targeted region and mutation type. Therefore, it remains unclear exactly how HacA functions.

Response: As suggested, the new title is “The Aspergillus flavus hacA gene in the unfolded protein response pathway is a candidate target for host-induced gene silencing.”

Furthermore, in this type of study, it is important to include a description of the knockout phenotypes (at least partially done according to Figure 5, but not addressed in the manuscript).

Response: As commented, a section describing null mutants of hacA in yeast and fungi has been added (lines 61-71).

Lastly, is recommended tuning down the relevance of the RNAi method because the survival rate of the transformants is still relatively high (implying limited efficacy in the silencing).

Response: Practically speaking, RNAi is a knock-down and not a knockout technique, and its effects are temporary. The integration locations in the genome and temporal expression of the vectors certainly affect the levels of miRNA production. This explanation, along with citations, has been added to the discussion (lines 527-529).

Introduction

The first paragraph explaining the mechanism of expression and regulation of hacA is difficult to follow. I strongly recommend rewrite ng it for clarity.

Response: Thank you for pointing this out. The statement was Indeed incomplete. The added text, including citations describing the mechanism, now reads– “These transcription factors bind to a consensus UPR element in the promoters of ER-stress responsive genes to activate their expression [5,6].”

Some specifics observations on this section: Line 31: “Endoplasmic” should be changed to “The endoplasmic reticulum”. Also, consider rephrasing this sentence because, although it is not incorrect, it reads awkwardly (folding is part of the maturation process, and translation of mRNA is a crucial event that happens at the ER). Line 42: Specify the species in which these experiments have been conducted to better understand the novelty of your results. Line 49: “(Saloheimo, 2003)” should be changed to the correct refence format. Line 53: “(2022)” should be completed with the right reference format. Line 61: “Research on Cance2” -> “Research on Cancer”. The use of commas would improve readability. Address the earlier comment about the hacA/hac1 naming convention.

Response: I apologized for the hasty preparation of the manuscript because of my upcoming retirement. Typos, grammatic errors, formatting issues have been carefully checked, and awkward sentences have been rewritten to enhance readability and smooth the flow. I hope the revised version meets the reviewer’s expectation.

Methods

The described experimental protocols are well-written. However, there are several fragments of the methodology embedded within the Result section, particularly in 3.6. Bioinformatics methods should be explained here too. Some of them are scattered within the in the Results section, particularly in sections 3.1 and 3.2.

Response: The method related text in section 3.6 (now section 3.5 due to the removal/merging of the prior section 3.2) has been deleted or merged into the relevant M&M section. The method text from the prior section 3.2 has been consolidated as a new section 2.2 (lines 154-164). Furthermore, sections 3.1 and 3.1 have been reorganized into a new section 3.1 (lines 248-276).

The RNA-seq data should be made available for consultation. The genomic information of the gene and the complete domain architecture of the protein are not mentioned.

Response: The research center is a US governmental facility, and its network is not accessible to the general public. In the “Data Availability Statement”, It states: Raw data and materials described in this study will be shared upon reasonable requests, in accordance with USDA policies and procedures.” If a researcher is interested in the stored RNA-seq data, our bioinformatician will be happy to provide it.

It would be very beneficial to the manuscript -and the reader- to include a diagram of the hacA gene, clearly illustrating the regions encoding each analyzed region.

Response: As commented, a diagram depicting the regions encoding basic/acidic amino acids, the basic leucine zipper, the conventional intron, and the non-conventional intron in now shown as Figure 1A

Results and conclusions

Subsection subtitles: Titles of this subsections should capture the main results; otherwise, they seem to refer more to methodological steps than outcomes. 3.1: Intron location and alignment with other species.

Response: The subtitles for the subsections have been revised to as follows:

3.1. The non-conventional intron in A. flavus hacA is 20 nucleotides long and its removal yields a truncated HacA lacking α-helical structures at the carboxyl terminal region

3.2. CRISPR/Cas9 mutagenesis of the hacA functional domain-coding regions and the non-conventional intron generates mutants exhibiting various phenotypes

3.3. The hacA mutants display different sensitivities to DTT

3.4. The hacA mutants all produce aspergillic acid and kojic acid but not anthraquinones or aflatoxin

3.5. The hacA RNAi affects regeneration and development of A. flavus protoplasts

Address the earlier comment about including genomic views and RNA-seq data. Figure 1: it would be important to show the position and alignment of the first 54 bp intron shared with the other species and the non-canonical 20 bp intron present in A. flavus, as this is central to argument in the text.

Response: The original Figure 1A is a snipped image from the JBrowse genome viewer. The top panel shows both strands of the non-conventional intron region and amino acid sequences translated from three reading frames, and it, in the bottom panel, shows paired-read reads from both ends (pink and blue lines). Excessive information may cause confusion to readers. The figure is somewhat redundant with Figure 1B, so it has been replaced with the diagram depicting features of the hacA gene suggested by reviewer. It should be noted that the 54-bp conventional intron and the non-conventional intron are located at different locations in the A. flavus hacA gene (please refer to the newly added Figure 1A and the revised text- lines 254-257). The non-conventional intron is not embedded within the first conventional intron.

Lines 217-218: Please complete/clarify the sentence. 3.2: Protein structure of hacA and hacAΔ Please clarify: The intron being analyzed is 20 bp long, but its removal results in a protein that is 93 amino acids shorter (438 vs. 345, according to lines 240 to 242). To lose 83 amino acids, a splicing event would have to remove 279 bp (93x3bp). The only explanation seems to be a frameshift that creates a premature stop codon. Please provide the explanation and evidence (e.g., alignment) in the manuscript.

Response: Yes. It is indeed the only reason. The removal of the 20-nucleotide non-conventional intron between nucleotides 701/706 and nucleotides 720/726 (flanked by a hexanucleotide direct re-peat, CTGCAG) results in a pre-termination codon, truncating the “wild-type” HacA. Please refer to Figure S3A for the wild-type and truncated HacA proteins.

Address the earlier comment about the bioinformatic methods and keep only the results. Please clarify: Why not assemble the hacA isoforms from the RNAseq data and translating them to amino acids instead of creating a HacAΔ manual sequence? How many isoforms are there?

Response: There is no need to generate the DNA sequence with the non-conventional intron removed by the suggested manner. Since the Nt20 sequence is known, a simple step deletion of these nucleotides following by using any sequence analysis software, will give the result.

Please clarify: The amino acid sequence of HacAΔ gives a result of 100% of identity in the AlphaFold database for A. flavus. Add the date of consultation of the 3D structure in the database.

Response: The hacA nucleotide sequences of A. flavus and A. oryzae are100% identical. The annotated proteins for them in the UniProKB database are based on different mRNA transcripts- the wild-type transcript for A. flavus HacA and the intron processed (removed) transcript for A. oryzae HacA. This allows the use of A. oryzae HacA as the proxy protein for the truncated A. flavus HacA.

The 3-D models generated by AlphaFold [38,39] (date of access: February 28, 2023) … (line 273)

3.3: Mutants’ genotypes and colony morphology as phenotype Figure 3: Add the wildtype phenotype as a first panel. Remove it from Fig 2B or explain its placement with the bZIP mutants.

Response: The CA14 (conidial) morphology was first mentioned in the M&M section- “Wild-type A. flavus CA14 (SRRC 1708), an L-morphotype isolate known for producing abundant spores ....” (lines 111-112).

To maintain consistency in presentation consistency, with four photos on each bottom of panels A, B and C, the wild-type photo thus was arranged this way. To clarify, the text “The wild-type (WT) CA14 strain is shown alongside the CS mutant for visual comparison and consistency in presentation” has been added to the legend of Figure 2B.

Figure 3: Add the length of the mutations with “bp” or “Kb” and note the location of the domains with “aa”.

Response: The legend in Figure 3 (now Figure 2) states “In the three panels, open triangles and solid inverted triangles followed by numbers indicate deleted and inserted nucleotides, respectively.” (lines 340-341). As requested, “aa” has been added to the respective WT sequences. 

Figure 3C: In the notation of each mutation and colony type, it would be advisable to specify the exact substitution to distinguish between the mutants that have the same annotation.

Response: Please refer to the WT amino acid sequence at the top. The mutated amino acids highlighted in yellow resulted from single nucleotide substitutions. The text “Only the predicted amino acid sequences before the removal of the respective introns in the mutants with single-nucleotide insertion or deletion are shown; the removal of these introns yields amino acid sequences identical to that of the truncated HacA∆.” (lines 336-338) has been added to the legend of Figure 2.

Section 3.4: resistance to ER stress Lines 313-314: Add reference for using DTT. Is there any way to verify that the UPRs response is activated by DTT in these colonies? Or a reference where it has been proven?

Response: DTT is commonly used in ER stress studies, and two references have been cited (line 350). Other agents, such as tunicamycin and brefeldin A, are also used for this purpose.

Line 319: Replace “It appeared that” might a more suitable statement as “We suspect” or “we hypothesize that”.

Response: The sentence has been revised to “The rich nutrient content of PDA appeared to exacerbate ER stress…”  (line 255).

Figure 4: Explain the notation X/Y/Z for each mutant in the caption. Address the ambiguity in the last two Nt20/down-arrow 1/Cs. Explicitly specify what Bsc, Nt20, and Zip are, and add the units in all mutations (bp, Kb) to maintain a consistent notation. Use a consistent notation for the bZIP domain (ZIP, zip, bZIP are found in the text).

Response: The text has been reviewed to ensure consistency in designations; now bZip is used throughout the text. The phrase “Open triangles and solid inverted triangles followed by numbers indicate deleted and inserted nucleotides, respectively.” has been repeated mentions in the legends of Figures 2, 3, and 4.

The designations of Bsc, bZip, and Nt20, having been repeatedly mentioned, should have been clear to the readers. As commented, the text “Bsc, bZip, and Nt20 are added to strain designations for reference to the mutated regions.” Has been included in the legend of Figure 3.

Section 3.5: secondary metabolites Please clarify: Why is the knockout not shown in the previous DTT experiments? It’s essential to understand this mutant, as well as the complete and combinatorial deletion of the three regions of interest.

Response: The KO denotes “the respective gene knockout mutants” as indicated in the figure legend.  The three KO mutants are unable to produce aspergillic acid, aflatoxin (and its precursors), and kojic acid, respectively, serving as negative controls on ADM, CAM, and KAM agar plates for these metabolites.

Figure 5: The results in this figure are very important. Thus, it would be important to enhance it by correcting the notation for the mutants. In the 7th column, the Nt20 mutant is missing the insertion length. Interestingly, this mutant has the most similar UV pattern to the WT. Could you discuss about it?

Response: Generally speaking, the conidial mutants including C and Cs (please refer to Figure 2A-C, 2B C(WT), and 2C-Cs for these morphologies) produce comparative amount of aflatoxin (Figure 5, TLC). Thus, only one strain is presented in Figure 4. The focus was on the sclerotial (S and Sc) and the mycelial (M) mutants.

Figure 6: Again, the results here are very interesting. They reinforce that the mycelial colonies with mutants in the bZIP domain or the 20 bp intron interferes with the synthesis of aflatoxins B1 and B2. Please discuss: Why do we see anthraquinones in the Bsc- mycelial mutants (Fig. 5, 4th and 5th columns) but there is no aflatoxin production, even though it is an intermediate in the synthesis pathway?

Response: The reasons for this observation are speculative and likely related to the mutated HacA proteins. As noted, “… how HacA, via the UPR pathway, controls the cascade regulation on secondary metabolite production remains to be understood. …the CA14 mycelial mutants likely produce altered conforms of HacA, either with or without an intact bZip domain, which is also true for the sclerotial mutants.” (lines 509-513)

Figure S5: Add panel B as a second panel of Figure 5.

Response: As suggested, the prior Figure S5 is now shown as Figure 5B.

Section 3.6: interference of HacA expression This section is reads more like methodology than results. Please be more direct in the results and shift the methods to the corresponding section.

Response: the methodology-related text has been deleted in the revised manuscript.

While it is true that there is a decrease of 10%-30% in the RNAi transformants, the survival rate is still high. Please discuss why gene silencing is so incomplete in this case. Emphasize why the RNAi was directed to the bZIP domain, particularly given the results of the previous experiments.

Response: “Practically speaking, RNAi is a knock-down and not a knockout technique, and its effects are temporary. The integration locations in the genome and temporal expression of the vectors certainly affect the levels of miRNA production” (lines 527-529). In contrast, CRISRP/Cas9 is a knockout technique, and the sequence lesions it creates are permanent, at the DNA level; thus, the mutants with lethal mutations did not survive.

Please consider another round of proofreading and editing.

Response: The manuscript has been extensively revised and edited. I hope the revised version meets the reviewer’s expectation.

Major comments

In this contribution, the authors aim to explore the function of the HacA protein in Aspergillus flavus by targeting three specific regions of its coding sequence (a basic amino acid/acid-enriched domain, the bZIP domain and a 20 bp intron) using CRISPR/Cas9 and RNAi gene silencing. The study reveals clear phenotypic differences, both morphological (colony types) and metabolic, particularly in the production of secondary metabolites of medical importance, such as aflatoxins. Hence, A. flavus hacA may be considered a good candidate for future host-induced gene silencing to control aflatoxin infection and contamination of crops and endanger human and animal health. However, before being accepted for publication in the Journal of Fungi it should address some opportunities for improvement to make the article more understandable.

 Detail comments

 Writing. While it is possible to follow the main ideas of the results, there are sections that are challenging to interpret for several reasons (including unfinished sentences, missing references, typographical errors, ambiguity of ideas, and lack of narrative flow). Therefore, is would strongly recommend a thorough round of proofreading and editing to improve clarity.

Response: The manuscript has been meticulously checked for the mentioned errors and corrected accordingly.

Figures. There should be a consistent and clear notation for all mutants across all figures and experiments. All the details of this notation should be clearly explained in the figure captions. Additionally, images of genomic views (such as read mappings) should show the genomic context (location and gene span) and be presented at a high resolution for clarity.

Response: Revision has been made to the figure legends, and the snipped RNA-seq image has been replaced with the new Figure 1A, as previously mentioned.

Reviewer 3 Report

In this study, Aspergillus flavus HacA functional domains were dissected by CRISPR/Cas9 to reveal the relationship between genotype and phenotype. The results showed that A. flavus hacA may be considered a candidate target for future host-induced gene silencing to control infection and aflatoxin contamination of crops. This design is reasonable, the data is enough, and the results are significant to the related readers. So my suggestion is acceptable after minor revision. 

Other comments were followed

1) Why the author selected hacA gene for study? Is it very important for A. flavus, or is it suitable for RNAi or gene knockout?

2) In  Figure 1, he author focused on identification of the non-conventional intron in A. flavushacA gene. why not focused on exson? Introns are always removed after transcripts become matured. 

3) In Figure 2, The 3-D models of A. flavus HacA (A) and HacA∆ (B) transcriptions factors were showed, what is the difference between these two version? can you get some functions difference from these structure differences?

4) In Figure 3, the author  constructed many mutants. the author should explain the purpose to create every mutant. Please explain in detail if possible.

5) DTT was used for stress in Figure 4, how about other stresses? such as H2O2, NaCl2. If the author have carried out other stresses?

6) In Figure 5, how many aspergillic acid, anthraquinones and kojic acid were produced by hacA mutants? can the author give the amounts of these metabolites?

7) In 3.6, Effect of hacA RNAi on regeneration and development of protoplasts was carried out. Can the author get some conclusion for constructing the RNAi? such as what is the best conditions for gene knockout in A. flavus?

Author Response

Reviewer 3

I thank the reviewer for the valuable comments.

Major comments

In this study, Aspergillus flavus HacA functional domains were dissected by CRISPR/Cas9 to reveal the relationship between genotype and phenotype. The results showed that A. flavus hacA may be considered a candidate target for future host-induced gene silencing to control infection and aflatoxin contamination of crops. This design is reasonable, the data is enough, and the results are significant to the related readers. So my suggestion is acceptable after minor revision. 

Response: Thank you. The manuscript has been extensively revised and edited.

Other comments were followed

1) Why the author selected hacA gene for study? Is it very important for A. flavus, or is it suitable for RNAi or gene knockout?

Response: The A. flavus hacA gene was first identified as a differentially expressed gene from in situ assays, where Zea mays kernels were inoculated with A. flavus strains (Gilbert et al. Core transcription factors affecting virulence in Aspergillus flavus during infection of maize. J. Fungi 2023, 9, 118). This work is a continuation of research aimed at identifying genes critical for growth and development of A. flavus as potential targets by HIGS.

2) In Figure 1, the author focused on identification of the non-conventional intron in A. flavus hacA gene. why not focused on exson? Introns are always removed after transcripts become matured. 

Response: Conventional introns are characterized by the consensus splicing sequence "GU" at the 5' splice site and "AG" at the 3' splice site, which are processed by the spliceosomal mechanism. The intron studied here is notn-conventional and supposedly is cleaved by a kinase-nuclease protein (lines 52 and 58). This non-conventional intron is processed under endoplasmic reticulum (ER)-stress.

3) In Figure 2, The 3-D models of A. flavus HacA (A) and HacA∆ (B) transcriptions factors were showed, what is the difference between these two version? can you get some functions difference from these structure differences?

Response: The two 3-D models represent the wild-type HacA (with the intact non-conventional intron) and the truncated HacA (the non-conventional intron removed), respectively. The wild-type HacA features a region with extensive α-helical structures from reside 245 to residue 410, whereas these α-helical structures are nearly absent in the carboxyl terminal portion of the truncated HacA, leaving only a small residual portion. Transcriptional factors, such as bZIP- and C6-type proteins, possess a DNA-binding domain in the amino terminal portion and a transcriptional activation domain (TA) in the carboxyl terminal portion. The removal of the non-conventional intron likely results in the production of an HacA protein with an active TA domain.

4) In Figure 3, the author constructed many mutants. the author should explain the purpose to create every mutant. Please explain in detail if possible.

Response: One objectives of this study is to expand the utility of the CRISPR/Cas9 technique by generating random mutations in the hacA gene to explore the relationship between genotype and phenotype. The mutations in the hacA sequence are purely random and cannot be predicted. They are the excision and repair of the targeted DNA sequence. The identified small changes, determined through DNA sequencing, are then correlated with different phenotypes. The “constructed” mutants are actually random mutants generated, which illustrates how subtle changes in a gene sequence can have pleotropic effects – insights not easily achieved by conventional gene deletion or disruption methods.

5) DTT was used for stress in Figure 4, how about other stresses? such as H2O2, NaCl2. If the author have carried out other stresses?

Response: The unfolded protein response (UPR) is caused by the accumulation of misfolded proteins in the endoplasmic reticulum (ER). DDT, a reducing agent, can break disulfide bonds in protein to cause ER stress and triggers the UPR. Other agents such as tunicamycin and brefeldin A also can be used. H2O2 and NaCl both affect redox homeostasis and have effects on the UPR. This has been reported in the most recent article by Yu et al.- HacA, a key transcription factor for the unfolded protein response, is required for fungal development, aflatoxin biosynthesis and pathogenicity of Aspergillus flavus (in Figure S4). Int J Food Microbiol. 2024;417:110693. Given that the effects of H2O2 and NaCl on A. flavus hacA have been reported, we directed our focus to studying secondary metabolite production.

6) In Figure 5, how many aspergillic acid, anthraquinones and kojic acid were produced by hacA mutants? can the author give the amounts of these metabolites?

Response: Due to resource limitations, only qualitative analyses were performed to determine the production of these metabolites. Nonetheless, the extent of pigmentation observed provides some insight into how their production were affected in these mutants.

7) In 3.6, Effect of hacA RNAi on regeneration and development of protoplasts was carried out. Can the author get some conclusion for constructing the RNAi? such as what is the best conditions for gene knockout in A. flavus?

Response: Literature indicates that different loop sizes have been used in the construction of RNAi vectors. It appears that the loop size may not be critical, but further systematic analyses are definitely needed to support this proposition.

Round 2

Reviewer 1 Report

Paper is significantly improved

Abstract

Certainly improved

“Besides wild type…”  change to “In addition to the wild type-like conidial morphology, …“

Don’t note the decreased survival rates twice  - take out the “In the Bsc and bZIP experimental sets…” sentence.

Add a sentence or two before the last sentence to explain why you can go from reduced transformation frequencies to a potential treatment strategy – this needs to be clearly laid out.

Introduction

Line 178/179  treatment duplicated

Line 213 “… mutants commonly exhibit …”  change to “… mutants not involving a targeted repair sequence commonly…”

Materials and Methods

Line 498 dimensional 

Results

Flow definitely improved

Line 746 reported for the hacA genes

Line 744  while due to the frameshift caused….the predicted HacA, designated HacAdelta, is truncated and consists of 345…

Line 863  (i) the wild-type conidial form (designated as C)…

Discussion

Line 1558 …significantly affect cellular growth…

Author Response

I thank Reviewer 1 for the comments and the corrections.

Abstract

“Besides wild type…”  change to “In addition to the wild type-like conidial morphology, …“

Response: As suggested, the sentence has been revised. (line 18)

Don’t note the decreased survival rates twice - take out the “In the Bsc and bZIP experimental sets…” sentence.

Response: As commented, the original sentence “the total number of transformants reached only 16% of the controls.“ has been removed. It now reads- “In the Bsc and bZip experimental sets, all observed mutations in both coding sequences were in-frame, suggesting that….” (lines 15-16)

Add a sentence or two before the last sentence to explain why you can go from reduced transformation frequencies to a potential treatment strategy – this needs to be clearly laid out.

Response: As suggested, the sentence “Defects in the hacA gene at the DNA and transcript levels affected the survival, growth, and development of A. flavus.” has been added before the concluding statement (Thus, this gene...). (lines 25-26)

Introduction

Line 178/179  treatment duplicated

Response: The duplicated words have been deleted. (line 60)

Line 213 “… mutants commonly exhibit …”  change to “… mutants not involving a targeted repair sequence commonly…”

Response: The suggested text has been added. (line 95)

Materials and Methods

Line 498 dimensional 

Response: The typo “dimentional” has been corrected. (line 156)

Results

Line 746 reported for the hacA genes

Response: The word “in” has been changed to “for”. (line 262)

Line 744  while due to the frameshift caused….the predicted HacA, designated HacAdelta, is truncated and consists of 345…

Response: As suggested, the sentence has been restructured. (lines 271-272)

Line 863  (i) the wild-type conidial form (designated as C)…

Response: The word “wild-type” has been added. (line 313)

Discussion

Line 1558 …significantly affect cellular growth…

Response: The word “cellular” has been used to replace “its”. (line 449)

Reviewer 2 Report

The author successfully addressed most of the recommendations made in the previous review and significantly improved the manuscript. However, I again strongly recommend him to further clarify and discuss the importance of RNAi experiments.

The author successfully addressed most of the recommendations made in the previous review and significantly improved the manuscript. However, I again strongly recommend him to further clarify and discuss the importance of RNAi experiments. For while CRISPR experiments demonstrate that gene disruption results in different phenotypes - some of great medical and agronomic importance - RNAi experiments do not conclusively show that gene silencing would produce the same phenotypic effects. Therefore, arguments should be made to show that silencing that genotype is involved in stress survival and aflatoxin production capacity.

Author Response

Reviewer 2

I thank Reviewer 2 for the comments.

Minor editing of English language required.

Response: The manuscript has been checked again and correct accordingly (another reviewer also pointed out specific text that needed to be revised).

The author successfully addressed most of the recommendations made in the previous review and significantly improved the manuscript. However, I again strongly recommend him to further clarify and discuss the importance of RNAi experiments. For while CRISPR experiments demonstrate that gene disruption results in different phenotypes - some of great medical and agronomic importance - RNAi experiments do not conclusively show that gene silencing would produce the same phenotypic effects. Therefore, arguments should be made to show that silencing that genotype is involved in stress survival and aflatoxin production capacity.

Response: The ultimate goal of A. flavus gene function studies over the past three decades has been to develop practical approaches to control aflatoxin contamination in crops, which significantly impacts the global economy and the health of humans and animals. As stated in the previous revision that “RNAi is a knock-down and not a knockout technique, and its effects are temporary.” Any emerging technology will always have its limitations, and it is important to remember that there is no magic bullet that can solve this global aflatoxin contamination problem once and for all. This hacA work, along with numerous genes involved in the growth, development, and pathogenicity of A. flavus studied by other researchers, is preliminary. Its purpose is to provide an additional candidate gene for the RNAi toolbox. The efficacy of this gene in combating A. flavus infection will have to be tested in a transgenic host plant (such as maize, cotton, or peanut), first in the green house and then in the field. There is still much work to be done, particularly in ensuring that an RNAi construct can be consistently and reliably expressed in a host plant. This will require using a tissue-specific and/or constitutive promoter to drive its expression, thereby maintaining effective levels of miRNAs to exert their antifungal capacity by degrading the hacA transcripts in A. flavus. Furthermore, even if sufficient levels of miRNAs are achieved, the effectiveness of these miRNAs in entering fungal cells remains a challenge. It should be noted that the RNAi work performed on A. flavus is not aimed to creating mutants with phenotypic changes similar to those obtained in the CRISPR experiments (though a small portion of the RNAi transformants did display phenotypic variations). Instead, the goal is to demonstrate a reduction in the survival of A. flavus. Conceptually, this implies that the binding of miRNAs to hacA transcripts in A. flavus (and potentially in crops in future efforts) would lead to partial or complete degradation of the bound hacA transcripts, resulting in both lethal and sublethal effects. If translated, the partial hacA transcripts would yield truncated HacA proteins, similar to those produced in the non-surviving transformants in the CRISPR experiments, which likely resulted from various out-of-frame mutations in the hacA gene. We can anticipate the adverse consequence of these conformation-changed, function-altered HacA proteins- whether derived from the out-of-frame mutated hacA sequences or from the degraded transcripts- on the UPR pathway. Regardless of whether the sequence change occur at the DNA or at transcript level, the outcome likely is the same: the accumulation of unfolded or misfolded proteins would severely disrupt the ER protein homeostasis, leading to toxic and lethal consequence for the cells. Conversely, the surviving transformants from the CRISPR experiments, which targeted the sequence region encompassing the basic and acidic amino acids (Bsc) and the basic leucine zipper (bZip), all have in-frame mutations. Notably, the resulting HacA proteins are not “truncated” but retain nearly the entire amino acid sequence of the wild-type HacA, aside from deletions of a few amino acid residues. The phenotypic changes are confined to the small populations of the surviving Bsc and bZip transformants. In summary, to be an effective control strategy, we cannot rely solely on a single-gene RNAi approach. A multiplex RNAi design that targets several genes critical to growth, development (conidiation and sclerotial formation), and aflatoxin production (over 20 genes in the aflatoxin biosynthesis pathway have been characterized) must be employed to achieve the desired outcomes. To accommodate the comment, I have added the following text to the discussion.

“…The partially degraded hacA transcripts likely yield truncated HacA proteins, similar to those produced in the non-surviving transformants from the CRISPR experiments, which presumably contained a hacA gene with various out-of-frame mutations. We can anticipate the adverse effects of these conformation-changed, function-altered HacA proteins on the UPR pathway. Regardless of whether the sequence changes occur at the DNA or transcript level, the outcome remains the same: the accumulation of unfolded or misfolded proteins would severely disrupt ER protein homeostasis, leading to toxic and lethal consequences for the cells.” (lines 534-541)

“…including designing a multiplex construct targeting genes critical to fungal growth, development (such as conidiation and sclerotial formation), and aflatoxin biosynthesis would be a practical and effective strategy.” (lines 544-547)